# High-Frequency Square Wave Injection Sensorless Control Method of IPMSM Based on Oversampling Scheme

Zhiqiang Wang [1,*], Qi Guo [1], Jifeng Xiao [2], Te Liang [2], Zhichen Lin [3] and Wei Chen [1]

1   School of Electrical Engineering, Tiangong University, Tianjin 300387, China
2   Baodi Power Supply Branch of State Grid Tianjin Electric Power Company, Tianjin 301800, China
3   College of Electrical Engineering, Zhejiang University, Hangzhou 310058, China
*   Correspondence: wangzhiqiang@tiangong.edu.cn

**Abstract:** In view of the disadvantages of the traditional high-frequency square wave signal injection method in the low-speed operation of high-power interior permanent magnet synchronous motor (IPMSM), such as the large error of rotor position calculation and delay of position update, a method based on high-frequency square wave signal injection is proposed to obtain an effective vector action current through oversampling. When the vector is zero, the current changes to not zero, but when the vector is effective, the current changes greatly. In the traditional sampling and calculation methods, the change of the zero-vector is ignored, resulting in errors, especially in the case of small power, and the errors are more obvious. Through the method of oversampling the current of the effective vector, the high-frequency response current of the effective vector is obtained. Through the reasonable demodulation method, the high-frequency response current of the effective vector is extracted, and then the rotor position information is obtained through the phase-locked loop. On this basis, the influence of the inherent nonlinear characteristics of the motor system and the sampling delay on the calculation of the rotor position is analyzed, and the error is compensated to obtain a more accurate rotor position.

**Keywords:** IPMSM; square wave injection sensorless control; current oversample

## 1. Introduction

As an energy-saving and environmentally friendly new energy product, electric vehicles have developed rapidly around the world in recent years [1–3]. The permanent magnet synchronous motor (PMSM) has been proven to be suitable for electric vehicle drive systems that require high torque density, high precision, and a wide speed range. In the process of controlling a permanent magnet synchronous motor, the rotor position and speed information obtained by the position sensor is usually needed to calculate the out-put voltage vector. However, due to the complex electromagnetic environment of electric vehicles and harsh operating environments, the position sensor is prone to failure and lose of position signals [4–6]. In order to improve the reliability of electric vehicle operations, the research on the sensorless control methods of high-performance permanent magnet synchronous motors has become a hot issue in recent years [7–9].

The speed sensorless control in the middle and high-speed range is realized by using the voltage model combined with the respective observers [10–12]. When the PMSM is at the low-speed stage, it is difficult to obtain the position information by using the observer because of the small amplitude of the back EMF [13–15]. At present, the main solution is to get the rotor position information by injecting a high-frequency signal and then designing a filter or a high-frequency current solution method, based on the salient characteristics of the IPMSM [16–32]. According to the different injection signal forms, high-frequency signal injection methods can be divided into the rotating sinusoidal injection method, the pulse vibration sinusoidal injection method, and the pulse vibration square wave injection

method. The use of a filter often causes a large delay during the update of the position information, which reduces the dynamic response performance of the current loop control system. The estimation results are easily affected by the nonlinear characteristics of the inverter and the cross-saturation of the inductors.

The main influence of the nonlinear factor of the inverter is the influence of the dead-time voltage. Reference [20] used the positive sequence component of the current in the rotating voltage injection method to compensate for the influence of the dead-time of the inverter on the negative sequence component and made up for the influence of the dead-time voltage on the high-frequency signal voltage. References [21–23] propose a reduced-order extended Kalman filter for filtering and position detection. This method takes into account factors such as modeling error and sampling noise and improves the anti-interference ability of the system. Using the square wave signal injection method effectively solves the influence of the filter. In order to solve the influence of inverter nonlinearity, references [24,25] analyze the influence of inverter nonlinearity on the high-frequency injection signal and the position detection and proposed a compensation strategy and a rotor/flux position estimation method based on frequency tracking to reduce the estimation error caused by the inverter nonlinearity. To solve the influence of cross saturation, the demodulation method proposed in reference [26] eliminates the low-pass filter by discretizing the current in the measurement frame. When the high-frequency square wave voltage injection scheme is used, the cross-saturation effect should be considered, and a direct compensation strategy is proposed to reduce the influence of cross-saturation on position estimation.

In addition, researchers also adopted the method of injecting the high-frequency square wave signal into the d-axis. There are many current demodulation methods proposed to extract the high-frequency response current, avoiding the use of filters. So, the square wave injecting method increases the accuracy of position calculation and improves the dynamic response performance of the system. When using this method to analyze and demodulate the high-frequency response current, the current changes in each control cycle are always approximately linear, that is, $\mathrm{d}i/\mathrm{d}t = \Delta i/\Delta t$. However, in the seven-segment SVPWM modulation, the zero-vector phase currents flowing in the inverter and motor system under the action of back EMF are not invariant but decaying gradually. The current rate of change in the zero-vector phase is not equal to that in the effective vector phase. In the traditional method and sampling scheme, the current during the action time of the zero-vector is always considered to be invariant, and the accuracy of the sampling currents will have large errors that affect the position estimation accuracy. In addition, the traditional method usually needs three cycles to update the position angle once, resulting in a delay in the position update.

In reference [27], the current scaling error, bias current, and error of A/D sampling in the motor control system are analyzed quantitatively, and the influence of the current measurement error on the accuracy of rotor position estimation is analyzed by taking the high-frequency injection method for IPMSM as an example. In addition, researchers also introduced compensation ideal to improve the accuracy of position angle estimation. For example, a three-dimensional compensation table was designed based on the relationship between the real rotor position and the convergence position [28], and a double filter compensation method was designed to eliminate harmonic errors [29].

This paper focuses on the research of the low-speed sensorless control of IPMSM and proposes a high-frequency square wave signal injection sensorless control method based on the current oversampling scheme. This method also injects a high-frequency square wave signal into the stator d-axis. Through current sampling at the beginning and end of the zero-vector and effective vector, respectively, in one PWM carrier cycle, more current change information generated by high-frequency signal injection is obtained. Then, the responding current signal demodulation method is used to extract the high-frequency response current when the effective vector acts, and the current signal is used to get more accurate rotor position information through the phase-locked loop. In addition, the proposed method samples multiple times in one PWM cycle, which improves the update

frequency of the position angle and the dynamic response performance of the system. Table 1 is the improvement of the proposed method.

**Table 1.** Improvement of the proposed method.

|  | Sample Mode | Weakness and Strength |
|---|---|---|
| The Traditional Method | Sampling three times in three PWM cycles, which is at the start time of the injection cycle, the middle time of the injection cycle, and the end time of the injection cycle | 1. The current sampling error during the zero-vector action process is ignored.<br>2. The angle update frequency is low. |
| The Proposed Method | Sampling four times in two PWM cycles, which is at the beginning and end of the zero-vector and effective vector, respectively, in one PWM carrier cycle | 1. The proposed method considers the current sampling errors $\varepsilon_\alpha$ and $\varepsilon_\beta$.<br>2. The angle update frequency is improved. |

## 2. Traditional Sensorless Control of IPMSM Based on High-Frequency Square Wave Voltage Injection

### 2.1. High-Frequency Square Wave Signal Injection Principle

Figure 1 is the traditional sensorless control block diagram based on the high-frequency square wave signal injection method.

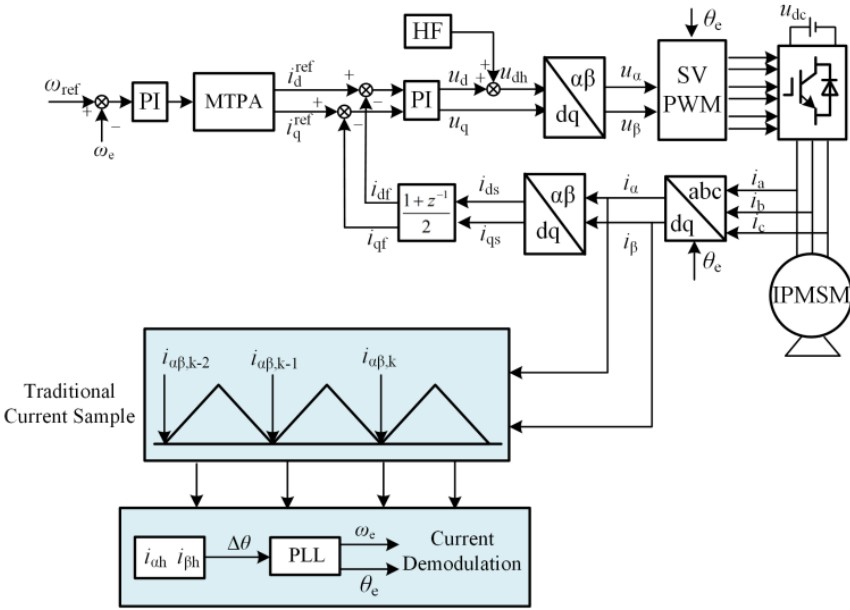

**Figure 1.** Block diagram of the traditional high-frequency signal injection method.

In Figure 2, d-q axis is the actual two-phase rotation coordinate system of the motor, and $d^e$-$q^e$ is the estimated two-phase rotation coordinate system calculated by the sensorless algorithm.

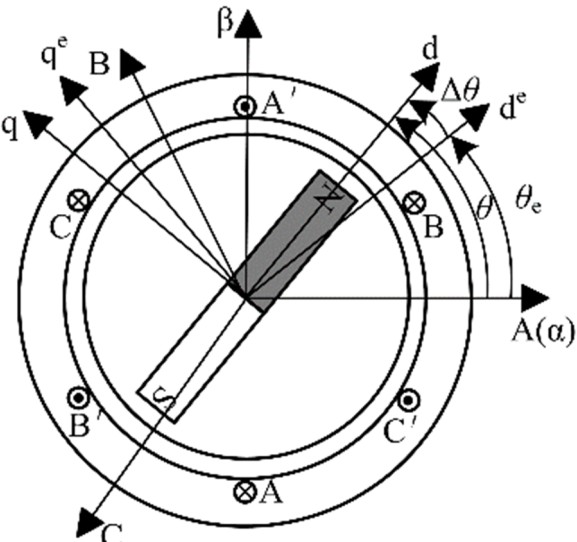

**Figure 2.** The relationship between different coordinate systems.

According to the coordinate relationship shown in Figure 2, the stator voltage equation of the motor can be obtained as follows:

$$
\begin{bmatrix} u_\mathrm{d} \\ u_\mathrm{q} \end{bmatrix} = R_\mathrm{s} \begin{bmatrix} i_\mathrm{ds} \\ i_\mathrm{qs} \end{bmatrix} + \begin{bmatrix} L_\mathrm{d} & 0 \\ 0 & L_\mathrm{q} \end{bmatrix} \frac{\mathrm{d}}{\mathrm{d}t} \begin{bmatrix} i_\mathrm{ds} \\ i_\mathrm{qs} \end{bmatrix}
$$
$$
+ \omega_\mathrm{e} \begin{bmatrix} 0 & -L_\mathrm{q} \\ L_\mathrm{d} & 0 \end{bmatrix} \begin{bmatrix} i_\mathrm{ds} \\ i_\mathrm{qs} \end{bmatrix} + \omega_\mathrm{e} \lambda_\mathrm{f} \begin{bmatrix} 0 \\ 1 \end{bmatrix} \tag{1}
$$

where, $u_\mathrm{d}$ and $u_\mathrm{q}$ are the reference voltages of $\mathrm{d^e}$-$\mathrm{q^e}$ axis, respectively; $R_\mathrm{s}$ is the stator resistance; $L_\mathrm{d}$ and $L_\mathrm{q}$ are the $\mathrm{d^e}$-$\mathrm{q^e}$ axis inductance, respectively; $i_\mathrm{ds}$ and $i_\mathrm{qs}$ are the currents of $\mathrm{d^e}$-$\mathrm{q^e}$ axis, respectively; and $\omega_\mathrm{e}$ is the estimated electric angular velocity of the motor.

The following formula can be obtained by transforming the voltage equation under the $\mathrm{d^e}$-$\mathrm{q^e}$ axis to the $\alpha$-$\beta$ axis through the coordinate transformation.

$$
\begin{bmatrix} u_\alpha \\ u_\beta \end{bmatrix} = R_\mathrm{s} \begin{bmatrix} i_\alpha \\ i_\beta \end{bmatrix} + 2L_2\omega_\mathrm{r} \begin{bmatrix} -\sin(2\theta_\mathrm{e}) & \cos(2\theta_\mathrm{e}) \\ \cos(2\theta_\mathrm{e}) & \sin(2\theta_\mathrm{e}) \end{bmatrix} \begin{bmatrix} i_\alpha \\ i_\beta \end{bmatrix}
$$
$$
+ \begin{bmatrix} L_1 + L_2\cos(2\theta_\mathrm{e}) & L_2\sin(2\theta_\mathrm{e}) \\ L_2\sin(2\theta_\mathrm{e}) & L_1 - L_2\cos(2\theta_\mathrm{e}) \end{bmatrix} \frac{\mathrm{d}}{\mathrm{d}t} \begin{bmatrix} i_\alpha \\ i_\beta \end{bmatrix} \tag{2}
$$
$$
+ \omega_\mathrm{r}\psi_\mathrm{f} \begin{bmatrix} -\sin(\theta_\mathrm{e}) \\ \cos(\theta_\mathrm{e}) \end{bmatrix}
$$

where, $u_\alpha$ and $u_\beta$ are voltage components in the $\alpha$-$\beta$ coordinate system; $i_\alpha$ and $i_\beta$ are current components in the $\alpha$-$\beta$ coordinate system, $L_1 = (L_\mathrm{d} + L_\mathrm{q})/2$, $L_2 = (L_\mathrm{d} - L_\mathrm{q})/2$; and $\theta_\mathrm{e}$ is the electric angle of the estimated pole position of the rotor.

Figure 3 shows the injected high-frequency square wave signal and carrier wave with the traditional method. From Figure 3, to the traditional high-frequency square wave signal injection sensorless control method, the injected square wave signal frequency is 1/2 of the carrier wave. To complete the demodulation process of the primary high-frequency current (i.e., to complete the primary position estimation), it is necessary to sample the current value at three points, that is, the current value at the beginning of each PWM cycle. As shown in Figure 3, *k*-2, *k*-1, and *k* are three sampling points. So, it requires at least three PWM control cycles to complete one position estimation.

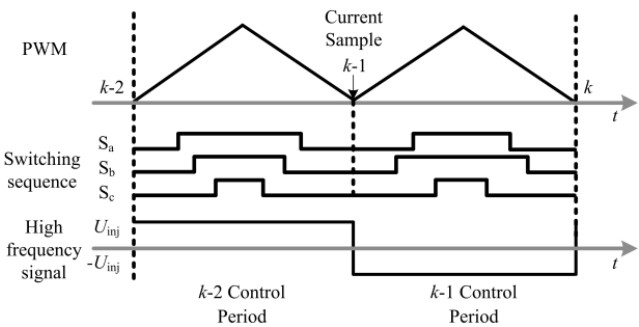

**Figure 3.** Traditional method of injecting signals and sampling points.

The frequency of the injected square wave signal is much higher than that of the fundamental wave signal. The high-frequency voltage equation of IPMSM in the α-β axis is:

$$\begin{bmatrix} u_{\alpha h} \\ u_{\beta h} \end{bmatrix} = \begin{bmatrix} L_1 + L_2\cos(2\theta_e) & L_2\sin(2\theta_e) \\ L_2\sin(2\theta_e) & L_1 - L_2\cos(2\theta_e) \end{bmatrix} \frac{\mathrm{d}}{\mathrm{d}t}\begin{bmatrix} i_{\alpha h} \\ i_{\beta h} \end{bmatrix} \tag{3}$$

where, $u_{\alpha h}$ and $u_{\beta h}$ are high-frequency voltage components in the α-β coordinate system, and $i_{\alpha h}$ and $i_{\beta h}$ are high-frequency current components in the α-β coordinate system.

The current differential expression is obtained from Equation (3)

$$\frac{\mathrm{d}}{\mathrm{d}t}\begin{bmatrix} i_{\alpha h} \\ i_{\beta h} \end{bmatrix} = \frac{1}{L_1^2 - L_2^2}\begin{bmatrix} L_1 - L_2\cos(2\theta_e) & -L_2\sin(2\theta_e) \\ -L_2\sin(2\theta_e) & L_1 + L_2\cos(2\theta_e) \end{bmatrix}\begin{bmatrix} u_{\alpha h} \\ u_{\beta h} \end{bmatrix} \tag{4}$$

At this time, the injected high-frequency square wave signal is as follows [6–8]:

$$\begin{bmatrix} u\mathrm{dh} \\ u\mathrm{qh} \end{bmatrix} = \begin{bmatrix} \pm U\mathrm{inj} \\ 0 \end{bmatrix} \tag{5}$$

Equation (5) is used to obtain the high-frequency signal under the α-β axis through coordinate transformation

$$\begin{bmatrix} u_{\alpha h} \\ u_{\beta h} \end{bmatrix} = \pm U_{\mathrm{inj}}\begin{bmatrix} \cos\theta_e \\ \sin\theta_e \end{bmatrix} \tag{6}$$

The voltage equation under the α-β axis can be obtained by taking Equation (6) into Equation (4) and simplifying

$$\frac{\mathrm{d}}{\mathrm{d}t}\begin{bmatrix} i_{\alpha h} \\ i_{\beta h} \end{bmatrix} = \frac{\pm U_{\mathrm{inj}}}{L_{\mathrm{dh}}}\begin{bmatrix} \cos\theta_e \\ \sin\theta_e \end{bmatrix} \tag{7}$$

where $L_{\mathrm{dh}}$ is the $d^e$-axis high-frequency inductive signal after high-frequency square wave voltage injection, approximately regarded as $L_{\mathrm{dh}} = L_{\mathrm{d}}$.

It is assumed that the current changes linearly in one sampling period [6–8]. $\mathrm{d}i/\mathrm{d}t$ being equal to $\Delta i/\Delta t$, Equation (7) can be arranged as follows:

$$\begin{bmatrix} \Delta i_{\alpha h} \\ \Delta i_{\beta h} \end{bmatrix} = \frac{\pm \Delta T U_{\mathrm{inj}}}{L_{\mathrm{dh}}}\begin{bmatrix} \cos\theta_e \\ \sin\theta_e \end{bmatrix} \tag{8}$$

where $\Delta i_{\alpha h}$ and $\Delta i_{\beta h}$ are the difference between the two sampling currents, and $\Delta T$ is the time difference of sampling.

The three-phase current values corresponding to three sampling points are transformed into a-β coordinate system:

$$\begin{cases} i_{\alpha\beta,k-2} = C_{3/2}i_{\mathrm{abc},k-2} \\ i_{\alpha\beta,k-1} = C_{3/2}i_{\mathrm{abc},k-1} \\ i_{\alpha\beta,k} = C_{3/2}i_{\mathrm{abc},k} \end{cases} \tag{9}$$

where $i_{\alpha\beta}$ is $[i_\alpha, i_\beta]$, and the subscripts *k*, *k*-1, and *k*-2, respectively, correspond to the three sampling times in Figure 3.

The above-obtained α-β axis currents include the basic frequency components and the high-frequency response components. Then, the basic frequency current and the high-frequency response current need to be demodulated.

When the injection signal is positive, the relationship between two adjacent sampling points is as follows:

$$\Delta i_{\alpha\beta k,k-1} = i_{\alpha\beta,k} - i_{\alpha\beta,k-1} = i_{\alpha\beta f} + \Delta i_{\alpha\beta h} \tag{10}$$

where $\Delta i_{\alpha\beta k,k-1}$ is the difference of the sampling current between two adjacent points, including the fundamental frequency current and the high-frequency response current.

When the injection signal is negative, the relationship between two adjacent sampling points is as follows:

$$\Delta i_{\alpha\beta k-1,k-2} = i_{\alpha\beta,k-1} - i_{\alpha\beta,k-2} = i_{\alpha\beta f} - \Delta i_{\alpha\beta h} \tag{11}$$

where $\Delta i_{\alpha\beta k-1,k-2}$ is the difference of the sampling current between two adjacent points, including the fundamental frequency current and the high-frequency response current.

The difference between Equations (10) and (11) can obtain the high frequency response current.

$$\Delta i_{\alpha\beta h} = \frac{1}{2}(\Delta i_{\alpha\beta k,k-1} - \Delta i_{\alpha\beta k-1,k-2}) \tag{12}$$

where $\Delta i_{\alpha\beta h}$ is the high-frequency response current under the α-β axis. $\Delta i_{\alpha\beta h}$ is substituted into Equation (8), and the rotor position angle can be calculated by the arctangent formula.

When the injection signal is negative, the relationship between two adjacent sampling points is as follows:

$$\theta e = \text{atan}(|\Delta i_{\beta h}|, |\Delta i_{\alpha h}|) \tag{13}$$

After the high-frequency response current is obtained, the rotor position can be calculated by substituting the high-frequency response current into Equation (13). Then, it is necessary to take the high-frequency response current as the input of the phase-locked loop (PLL). The rotor speed and position can be obtained by PLL shown in Figure 4.

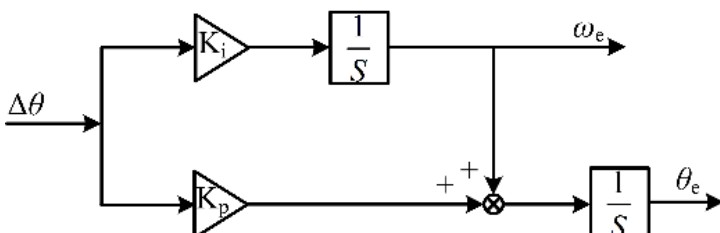

**Figure 4.** The principal block diagram of PLL.

PLL calculates the speed and rotor position by

$$\begin{cases} \omega_e = G_{\omega_e}^{PLL}(s)\Delta\theta \\ \theta_e = G_{\theta_e}^{PLL}(s)\Delta\theta \end{cases} \tag{14}$$

$$\begin{cases} G_{\omega_e}^{PLL}(s) = K_i \\ G_{\theta_e}^{PLL}(s) = K_P + \frac{K_i}{s} \end{cases} \tag{15}$$

where $G_{\omega_e}^{\mathrm{PLL}}$ is the speed transfer function in PLL, and $G_{\omega_e}^{\mathrm{PLL}}$ is the position angle transfer function in PLL.

$$
\begin{cases}
\mathrm{K_p} = \frac{w_c}{2}\sqrt{\frac{\tan^2\gamma}{1+\tan^2\gamma}} \\
\mathrm{K_i} = \frac{w_c^2}{2}\sqrt{\frac{1}{1+\tan^2\gamma}}
\end{cases}
\tag{16}
$$

where $w_c$ is the cut-off frequency of the system, $\gamma$ is the phase angle margin, and $\mathrm{K_p}$ and $\mathrm{K_i}$ are the proportional and integral coefficients in the PLL [21].

### 2.2. Error Factor Analysis of Traditional High Frequency Square Wave Signal Injection Method

### 2.2.1. The Ignored Current Sampling Error during the Zero-Vector Action Process

The square wave voltage was injected as shown in Figure 3 into d-axis. To the traditional high-frequency square wave injection method, considering that the current-loop control is digital, the sampling current of each PWM cycle is linear, so the current change during the zero-vector action stage is always ignored.

As shown in Figure 5, taking A-phase current as an example, in a PWM cycle, it is approximately considered that the current is an ideal change in the phase of zero-vector action, and the current value at zero time of each PWM cycle is equal to that at the beginning of the effective vector. In Figure 5, $i_{a,idc0}$, $i_{a,idc1}$, $i_{a,idc2}$, and $i_{a,idc3}$ are used to represent the three sampling times corresponding to the traditional method. The traditional method chooses to sample at the beginning of the PWM cycle. In fact, during the phase of the zero-vector action process, due to the action of back EMF and the continuous current of the diode, the current will not be changed. In Figure 5, $\Delta i_{a,act0}$ and $\Delta i_{a,act1}$ are the current change values during the zero-vector phase action process. Due to the current change in the zero-vector phase, the sampling error occurs at the beginning of zero-vector action time and at the beginning of effective vector action time in one PWM cycle. It can be seen from Equation (13) that the above sampling error will lead to the obtained high-frequency response current of the α-β axis being inaccurate. The sampling error of the high-frequency response current will cause errors in the final rotor position calculation.

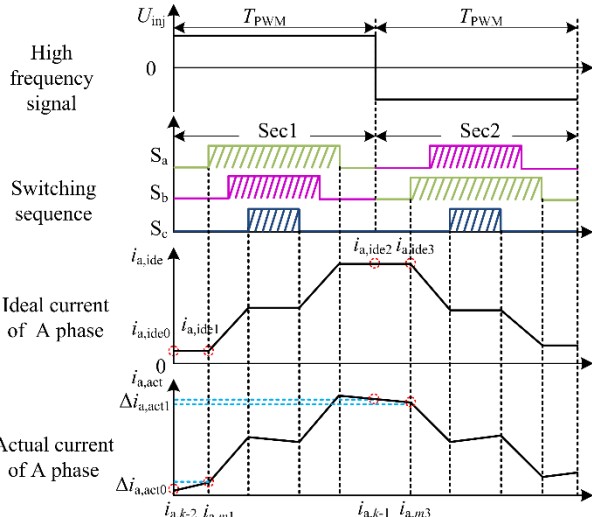

**Figure 5.** Changes of A-phase current with the traditional method.

From Figure 5, the error between the zero-time sampling current at the beginning of the PWM cycle and the sampling current at the beginning of the effective vector is

$$
\Delta i_{a,act0} = i_{a,m1} - i_{a,k-2}
\tag{17}
$$

Similarly, the error between the zero-time sampling current at the beginning of the PWM cycle and the sampling current at the beginning of the effective vector is

$$\Delta i_{\text{a,act1}} = i_{\text{a},m3} - i_{\text{a},k-1} \tag{18}$$

After the coordinate transformation of Equations (17) and (18), the current error under the α-β axis can be obtained, which is expressed as:

$$\begin{cases} \Delta i_{\alpha\beta0} = i_{\alpha\beta,m3} - i_{\alpha\beta,k-1} \\ \Delta i_{\alpha\beta1} = i_{\alpha\beta,m1} - i_{\alpha\beta,k-2} \end{cases} \tag{19}$$

In Equation (19), when calculating the rotor position angle by using the current difference value, an error will be generated. The expression of the error is:

$$\begin{cases} \varepsilon_\alpha = i_{\alpha,m1} - i_{\alpha,k-2} + i_{\alpha,m3} - i_{\alpha,k-1} \\ \varepsilon_\beta = i_{\beta,m1} - i_{\beta,k-2} + i_{\beta,m3} - i_{\beta,k-1} \end{cases} \tag{20}$$

The position angle position error $\Delta\theta_e$ caused by sampling error can be expressed as

$$\Delta\theta_e = \text{atan}(\varepsilon_\beta, \varepsilon_\alpha) \tag{21}$$

Therefore, when the current change during the zero-vector action process is ignored, there will be current errors, $\varepsilon_\alpha$ and $\varepsilon_\beta$, which will cause an error in the rotor position calculation.

### 2.2.2. The Low Angle Update Frequency and the Long Position Delay

To the traditional high-frequency square wave signal injection sensorless control algorithm, the frequency of the injected square wave signal is generally equal to 1/2 of the PWM cycle frequency. According to Equation (12), the traditional scheme needs three current samples to demodulate one high-frequency response current, so the angle update needs at least three PWM cycles. The traditional method program execution sequence diagram is shown in Figure 6. There are three sampling points: $k$-2, $k$-1, and $k$. After current sampling, the coordinate transformation is carried out to obtain $i_{\alpha\beta,k-2}$, $i_{\alpha\beta,k-1}$, and $i_{\alpha\beta,k}$. Then, current demodulation is carried out to obtain the high-frequency response current under the α-β axis. In the $k$ cycle, the motor speed and position information are obtained through PLL. Therefore, the traditional method needs three PWM cycles to calculate the rotor position, which will delay the position update for three cycles and cause the position angle error. The above phenomenon will be more serious when the speed is higher and the carrier ratio is relatively lower.

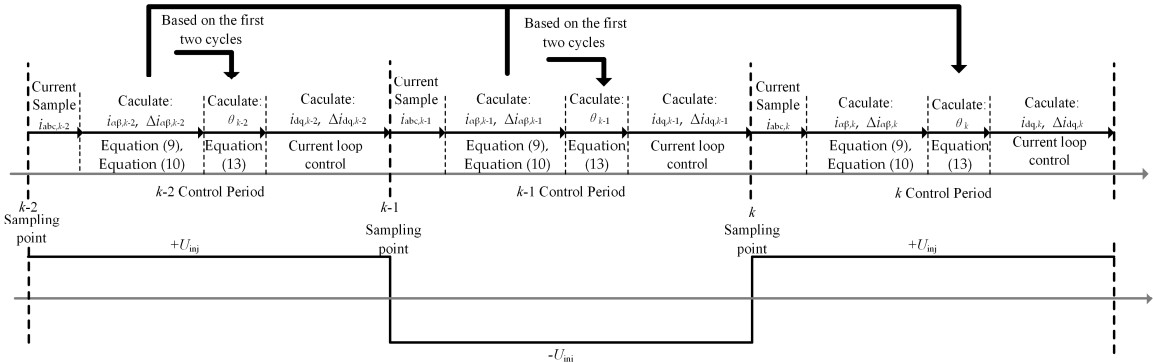

**Figure 6.** Program execution sequence diagram of the traditional method.

### 3. A Novel Sensorless Control Method of High-Frequency Square Wave Voltage Injection Based on Oversampling Scheme

*3.1. Oversampling and Effect Analysis of High-Frequency Response Current*

The high-frequency square wave signal injected by the novel method proposed in the paper is shown in Figure 7. Although the frequency of the signal is still half of the carrier frequency, in order to reduce the error of the estimated position angle, this paper adopts a current oversampling scheme to obtain a more accurate high-frequency response current.

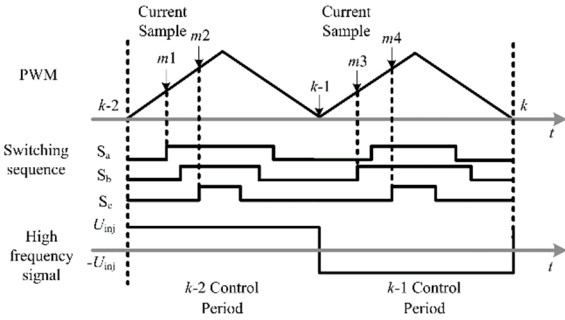

**Figure 7.** Proposed method current oversampling and injection signal waveform.

From Figure 7, when the injection signal is positive, two sampling currents are conducted, i.e., $m_1$, $m_2$ sampling points in the figure; when the injection signal is negative, two sampling currents are conducted, i.e., $m_3$, $m_4$ sampling points in the figure. Among them, $m_1$, $m_2$, $m_3$, and $m_4$ are the current sampling points used for calculating the position angle, respectively; $m_1$ and $m_2$ are the sampling points at the beginning and end of the effective vector of the first half period of $k$-2, and $m_3$ and $m_4$ are the sampling points at the beginning and end of the effective vector of the first half period of $k$-1. Compared with the traditional method of sampling three times in three PWM cycles, the proposed method samples four times in two PWM cycles. At the same time, $k$-2 and $k$-1 are sampling points for current loop control. Because the feedback current used in the current loop is still the zero-time sampling value of the PWM cycle, the control effect of the current loop is not affected.

Based on the oversampling scheme of stator current shown in Figure 7, the program execution block diagram of the oversampling method is shown in Figure 8.

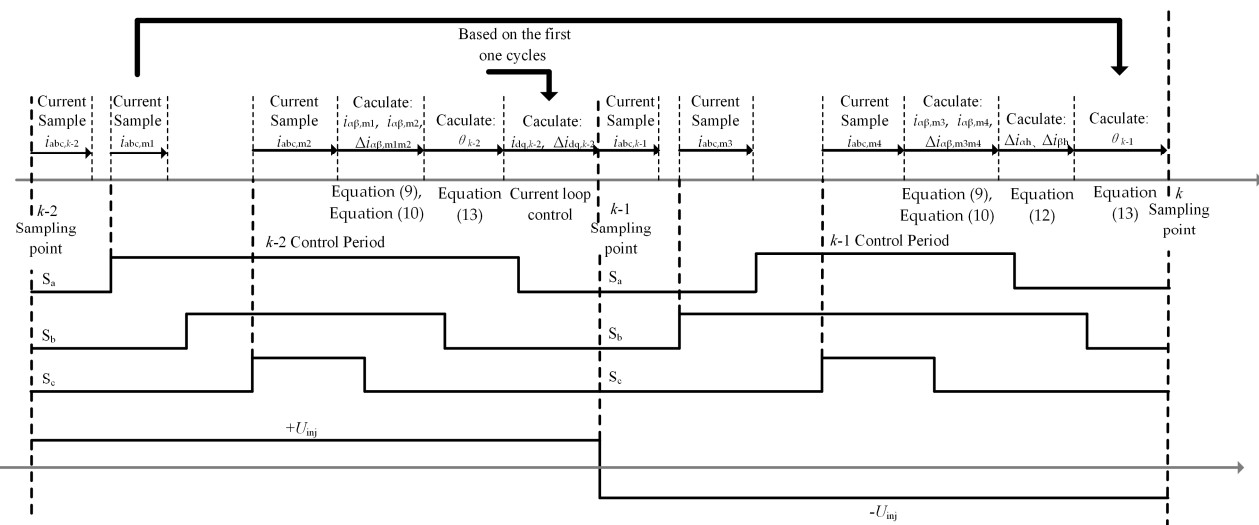

**Figure 8.** Program execution sequence diagram of the proposed method.

In the $k$-2 period, the oversampling currents are $i_{abc,m1}$ and $i_{abc,m2}$ at the beginning and end of the effective vector, respectively. Then, $i_{\alpha\beta,m1}$ and $i_{\alpha\beta,m2}$ are obtained by

coordinate transformation, and the current difference is calculated in the *k*-2 period to obtain $\Delta i_{\alpha\beta,\mathrm{m2m1}}$.

In the *k*-1 period, the oversampling current are $i_{\mathrm{abc,m3}}$ and $i_{\mathrm{abc,m4}}$ at the beginning and end of the effective vector, respectively, and then $i_{\alpha\beta,\mathrm{m3}}$ and $i_{\alpha\beta,\mathrm{m4}}$ are obtained by coordinate transformation. The current difference is calculated in the *k*-1 period, and $\Delta i_{\alpha\beta,\mathrm{m4m3}}$ is obtained. In the *k*-1 period, the high-frequency response current $\Delta i_{\alpha\beta h}$ is calculated, and the motor speed and position information are obtained by PLL. Therefore, the oversampling current method only needs two PWM cycles to get the rotor position. Compared with the traditional method, the oversampling current method can reduce the effect of position angle delay and increase the angle update frequency.

Because the high-frequency square wave signal is not injected into the inverter at the initial time of zero-vector action, there is no high-frequency response current in the zero-vector phase. To the traditional high-frequency square wave signal injection method, the current sampling point is usually selected at the zero-time of the PWM cycle for the convenience of sampling and calculation. In Figure 9, $i_{\mathrm{a},k-2}$, $i_{\mathrm{a},k-1}$, and $i_{\mathrm{a},k}$ cause the current change to produce errors. Section 2.2 has analyzed the reasons of the errors.

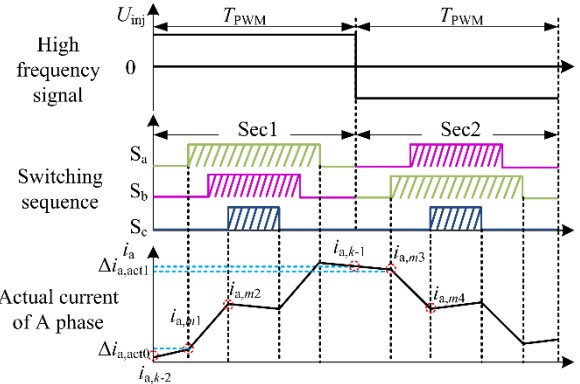

**Figure 9.** The actual change of phase A current with the proposed method.

The sampling points of the novel method ($i_{\mathrm{a,m1}}$, $i_{\mathrm{a,m2}}$, $i_{\mathrm{a,m3}}$, $i_{\mathrm{a,m4}}$) are selected at the beginning and end of the effective vector, respectively, as shown in Figure 9.

$\Delta i_{\mathrm{a,act0}}$ and $\Delta i_{\mathrm{a,act1}}$ are current errors calculated as follows:

$$\begin{cases} \Delta i_{\mathrm{a,act0}} = i_{\mathrm{a},k-2} - i_{\mathrm{a,m1}} \\ \Delta i_{\mathrm{a,act1}} = i_{\mathrm{a,m3}} - i_{\mathrm{a},k-1} \end{cases} \tag{22}$$

The novel method adopts the current oversampling scheme to sample at the beginning and end of the effective vector in a PWM cycle. When the high-frequency square wave voltage signal has been injected into the inverter, the high-frequency response current changes, and the linear change can be seen approximately. Therefore, the proposed method considers the current sampling errors $\varepsilon_\alpha$ and $\varepsilon_\beta$ and obtains more accurate current sampling results, which can improve the accuracy of the rotor position calculation.

### 3.2. The High-Frequency Response Current Demodulation with the Proposed Method

When the high-frequency square wave signal is injected, the oversampling currents include the high-frequency response components and the basic frequency components. The three-phase currents are transformed into the α-β coordinate system.

$$\begin{bmatrix} i_\alpha \\ i_\beta \end{bmatrix} = C_{3s/2s} \begin{bmatrix} i_{\mathrm{a}} \\ i_{\mathrm{b}} \\ i_{\mathrm{c}} \end{bmatrix} \tag{23}$$

The demodulation process of high-frequency response current is as follows:

$$
\begin{cases}
\Delta i_{\alpha,m2m1} = i_{\alpha,m2} - i_{\alpha,m1} = \Delta i_{\alpha h} + i_{\alpha f} \\
\Delta i_{\alpha,m4m3} = i_{\alpha,m4} - i_{\alpha,m3} = -\Delta i_{\alpha h} + i_{\alpha f}
\end{cases}
\tag{24}
$$

$$
\begin{cases}
\Delta i_{\beta,m2m1} = i_{\beta,m2} - i_{\beta,m1} = \Delta i_{\beta h} + i_{\beta f} \\
\Delta i_{\beta,m4m3} = i_{\beta,m4} - i_{\beta,m3} = -\Delta i_{\beta h} + i_{\beta f}
\end{cases}
\tag{25}
$$

As shown in Figure 8, $\Delta i_{\alpha,m2m1}$ and $\Delta i_{\beta,m2m1}$ are the differences between the two current samples in the α-β coordinate system when the square wave signal is positive. At this time, the high-frequency response current is positive, so it can also be expressed as the sum of the high-frequency response current and the basic frequency current. $\Delta i_{\alpha,m4m3}$ and $\Delta i_{\beta,m4m3}$ are the difference between two current samples when the square wave signal in the α-β coordinate system is negative. At this time, the high-frequency response current is negative, so it can also be expressed as the difference between the high-frequency response currents and the fundamental frequency currents. $\Delta i_{\alpha h}$ and $\Delta i_{\beta h}$ are the high frequency response currents, and $i_{\alpha f}$ and $i_{\beta f}$ are the fundamental frequency currents.

The high-frequency response current can be obtained by making difference between Equations (24) and (25), respectively.

$$
\Delta i_{\alpha h} = \frac{1}{2}(\Delta i_{\alpha,m2m1} - \Delta i_{\alpha,m4m3})
\tag{26}
$$

$$
\Delta i_{\beta h} = \frac{1}{2}(\Delta i_{\beta,m2m1} - \Delta i_{\beta,m4m3})
\tag{27}
$$

Cross vector multiplication by using Equations (26) and (27)

$$
\begin{aligned}
\sin \Delta\theta &= \frac{L_d}{\Delta T U_{inj}}[\Delta i_{\alpha h} \cdot (-\sin\theta_{e,k-1}) + \Delta i_{\beta h} \cdot \cos\theta_{e,k-1}] \\
&= \sin\theta_{e,k}\cos\theta_{e,k-1} - \cos\theta_{e,k}\sin\theta_{e,k-1}
\end{aligned}
\tag{28}
$$

When $\Delta\theta$ approaches 0, $\sin\Delta\theta \approx \Delta\theta$, so get the next formula,

$$
\begin{aligned}
\Delta\theta &= \frac{L_d}{\Delta T U_{inj}}[\Delta i_{\alpha h} \cdot (-\sin\theta_{e,k-1}) + \Delta i_{\beta h} \cdot \cos\theta_{e,k-1}] \\
&= \sin\theta_{e,k}\cos\theta_{e,k-1} - \cos\theta_{e,k}\sin\theta_{e,k-1}
\end{aligned}
\tag{29}
$$

Derived from the above formula, the position of current demodulation is shown in Figure 10.

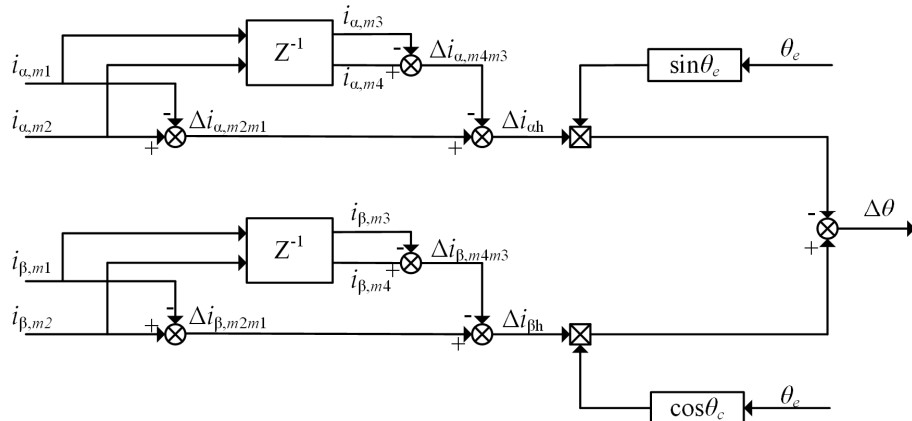

**Figure 10.** The current demodulation diagram of proposed method.

After obtaining $\Delta\theta$, take it as the input value of PLL, and acquire the rotor position and speed through PLL. The schematic diagram of PLL is shown in Figure 4 in Section 2.1. Figure 11 is the block diagram of the proposed method in the paper.

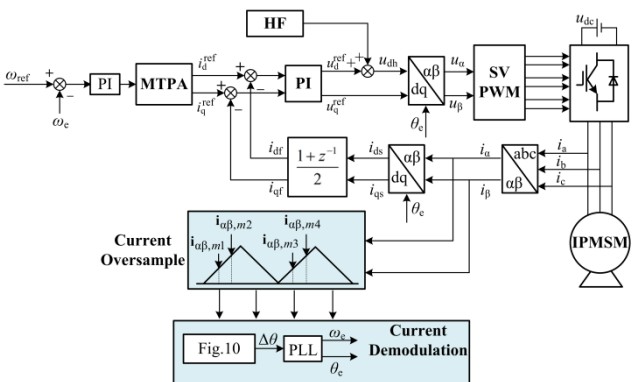

**Figure 11.** Block diagram of the proposed high-frequency signal injection method.

## 4. Experiment Result and Analysis

To verify the effectiveness of the proposed method in this paper, the PMSM experimental platform as shown in Figure 12 is set up. The experimental platform consists of the dynamometer, DC power supply, inverter, control circuit, and PMSM. The dynamometer is an asynchronous motor driven by the ABB ACS-800 AC drive. The digital signal processing chip of the control circuit is TMS320F28335, and the inverter adopts the electric vehicle GD12-WDI power unit produced by Semikron. The motor parameters are shown in Table 2. The amplitude of the high-frequency square wave voltage injected into the de axis is 40 V and the frequency is 2.5 kHz. The switching frequency of the experimental system is 5 kHz.

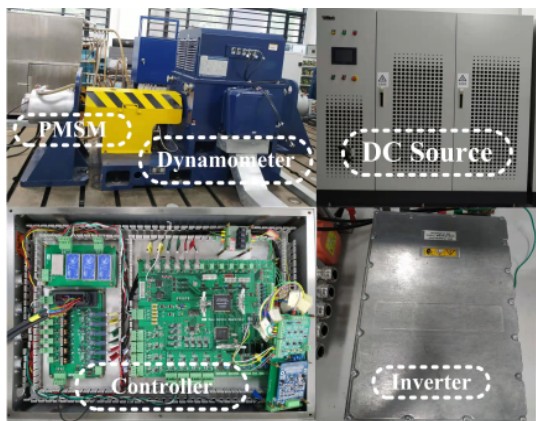

**Figure 12.** Photo of the experiment system.

**Table 2.** IPMSM parameter.

| Parameter | Value | Parameter | Value |
|---|---|---|---|
| Rates Power | 20 kW | Rated Speed | 3000 r/min |
| *d*-axis Inductance | 0.209 mH | Pole Pairs | 4 |
| *q*-axis Inductance | 0.333 mH | Resistance | 10.23 mΩ |
| Rated Torque | 64 Nm | Flux Linkage | 0.071 Wb |
| Rated Voltage | 300 V | Rated Current | 94 A |

### 4.1. Load Start and Brake Experiment at Low Speed

Figure 13 shows the experimental results of the load starting and braking at low speed with the traditional method and the proposed method, respectively. Due to the driving motor, electric vehicles always work under the torque control mode. So, during the experiment, the dynamometer motor was in speed mode control and the tested motor was in torque mode control. At first, the dynamometer system was given zero-speed instruction

to ensure that the dynamometer motor is static. Then the rotor position estimation program was started. The output torque of the tested motor was controlled from zero to the 1.5 times rated load (96 Nm). Then, the speed of the dynamometer motor was adjusted from 0 r/min to 400 r/min after 5 s. When the speed reached a steady state, the dynamometer motor speed continued to adjust to 0 r/min. The slope adjustment time of the dynamometer motor speed is about 5 s.

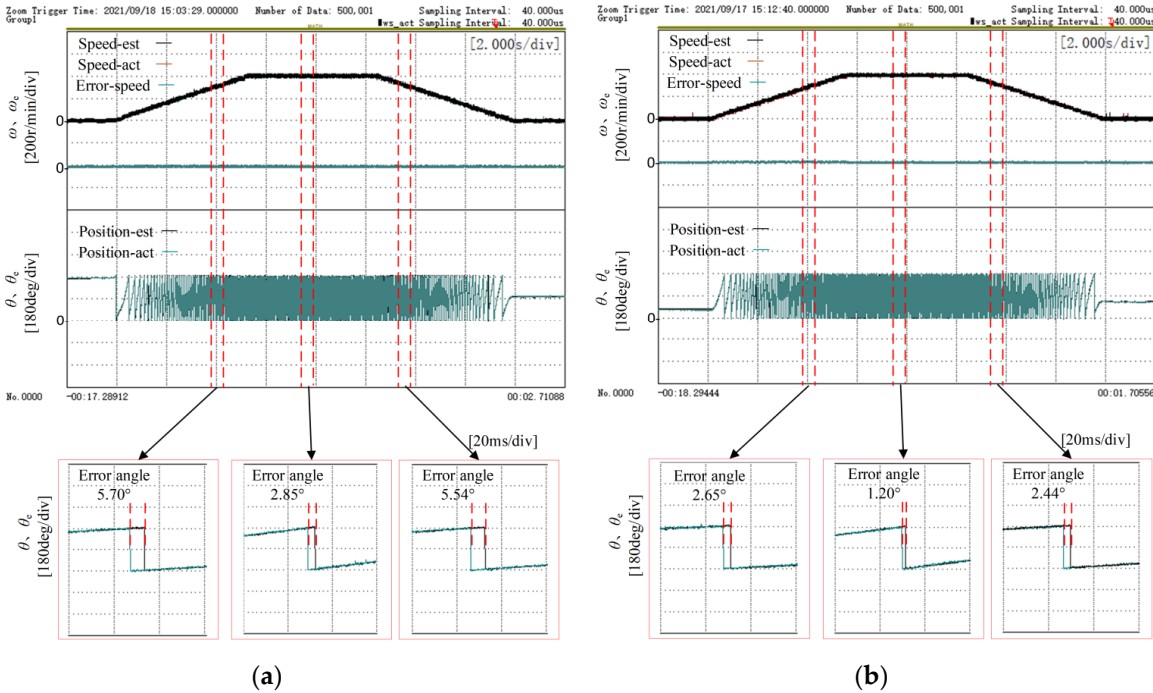

(**a**)                                    (**b**)

**Figure 13.** The speed control of dynamometer with traditional method and the proposed method from 0 r/min to 400 r/min, and from 400 r/min to 0 r/min: (**a**) traditional method; (**b**) the proposed method.

From the local enlarged view in Figure 13a, it can be clearly calculated that with the traditional method that the angle error is 5.70° during acceleration, the angle error is 2.85° when the speed of the dynamometer was controlled at 400 r/min, and the angle error is 5.54° during deceleration. From the local enlarged view in Figure 13b, it can be clearly calculated that with the proposed method in the paper, the angle error is 2.65° during acceleration, 1.20° when the speed of the dynamometer was controlled at 400 r/min and 2.44° during deceleration.

Figure 14 shows the similar experimental results. However, the speed of the dynamometer motor was adjusted from 0 r/min to 100 r/min and then returned to 0 r/min. From the local enlarged view in Figure 14a,b, it also can be seen that the proposed method in the paper has better estimation accuracy of electrical position angle of the rotor.

*4.2. Step Load Experiment at Constant Speed*

Figure 15 shows the experimental results of the step load with traditional method and proposed method, respectively. During the experiment, the dynamometer motor was in torque mode control and the tested motor was in speed mode control. At first, the tested rotor position estimation program was started. The output speed of the tested motor was controlled from zero to 400 r/min. After the tested motor speed reached the steady state, the step torque signal was given to the dynamometer system and the out-put torque of dynamometer motor was improved from zero to 30 Nm, 64 Nm (the rated load), and 96 Nm (the 1.5 times rated load). After the tested motor speed reached the steady state, the output

torque of dynamometer motor was reduced from 96 Nm to 64 Nm, 30 Nm, and 0 Nm, successively. The step adjustment time of the dynamometer motor torque is about 1 s.

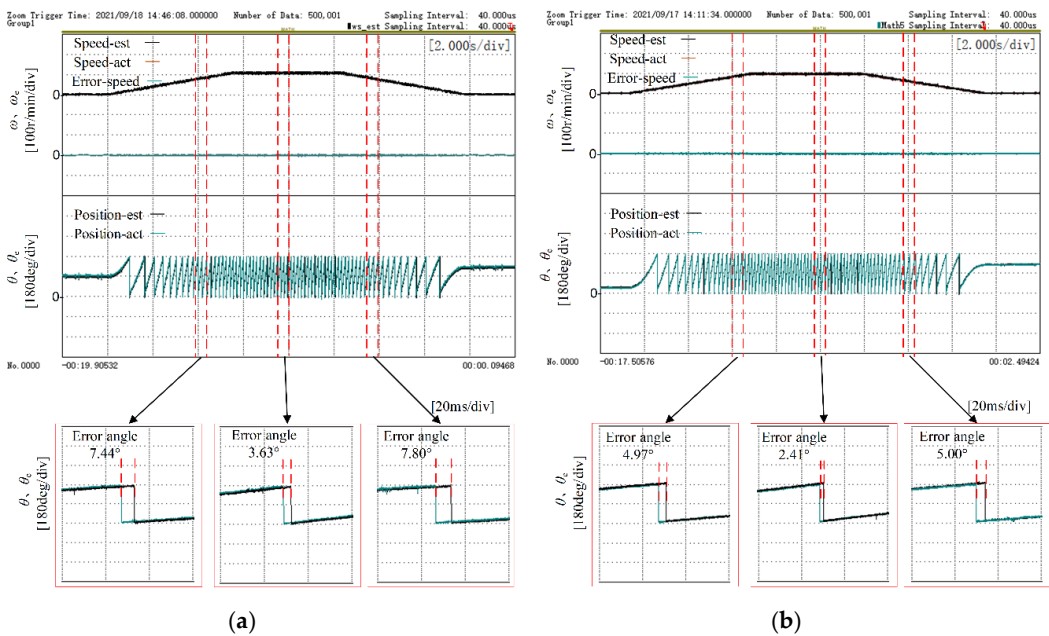

**Figure 14.** The speed control of dynamometer with traditional method and the proposed method from 0 r/min to 100 r/min, and from 100 r/min to 0 r/min: (**a**) traditional method; (**b**) the proposed method.

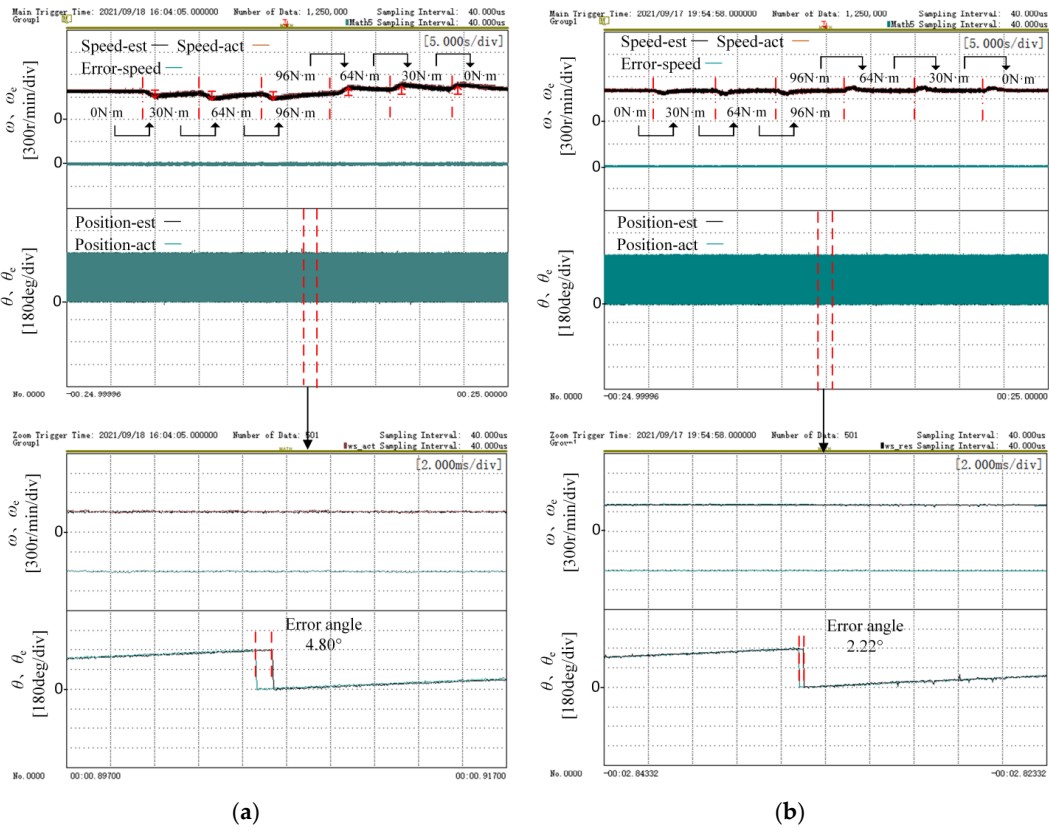

**Figure 15.** The speed control of tested motor with traditional method and proposed method at 400 r/min: (**a**) traditional method; (**b**) the proposed method.

From the local enlarged view in Figure 15a, it can be clearly calculated that with the traditional method, the angle error is 4.80° when the applied torque by dynamometer motor changed from the rated load of 64 Nm to the 1.5 times rated load of 96 Nm. From the local enlarged view in Figure 15b, it can be clearly calculated that with the proposed method, the angle error is 2.22° when the applied torque by dynamometer motor changed from the rated load of 64 Nm to the 1.5 times rated load of 96 Nm. Compared with the traditional method, the position angle error of the proposed method is smaller. As can be seen from the tested motor speed waveform in Figure 15a, when the applied torque by dynamometer motor changed abruptly, the speed of the traditional method changes greatly and recovers slowly. As can be seen from Figure 15b, when the torque applied by the dynamometer motor changed abruptly, the speed change of the proposed method was small, and it recovered quickly.

Figure 16 shows the similar experimental results. However, the output speed of the tested motor was controlled from zero to 100 r/min. Compared with the traditional method, the proposed method has a more accurate estimation of the position angle, and when the load changes suddenly, the deviation from the actual position angle is smaller. Therefore, the speed change is smaller and the speed recovers quickly after the change. The proposed method in this paper can be applied to the load step process at low speeds very well. Actually, the tested motor can operate at higher speeds with bigger loads with the proposed method. So, the proposed method can better meet the application requirements of electric vehicles.

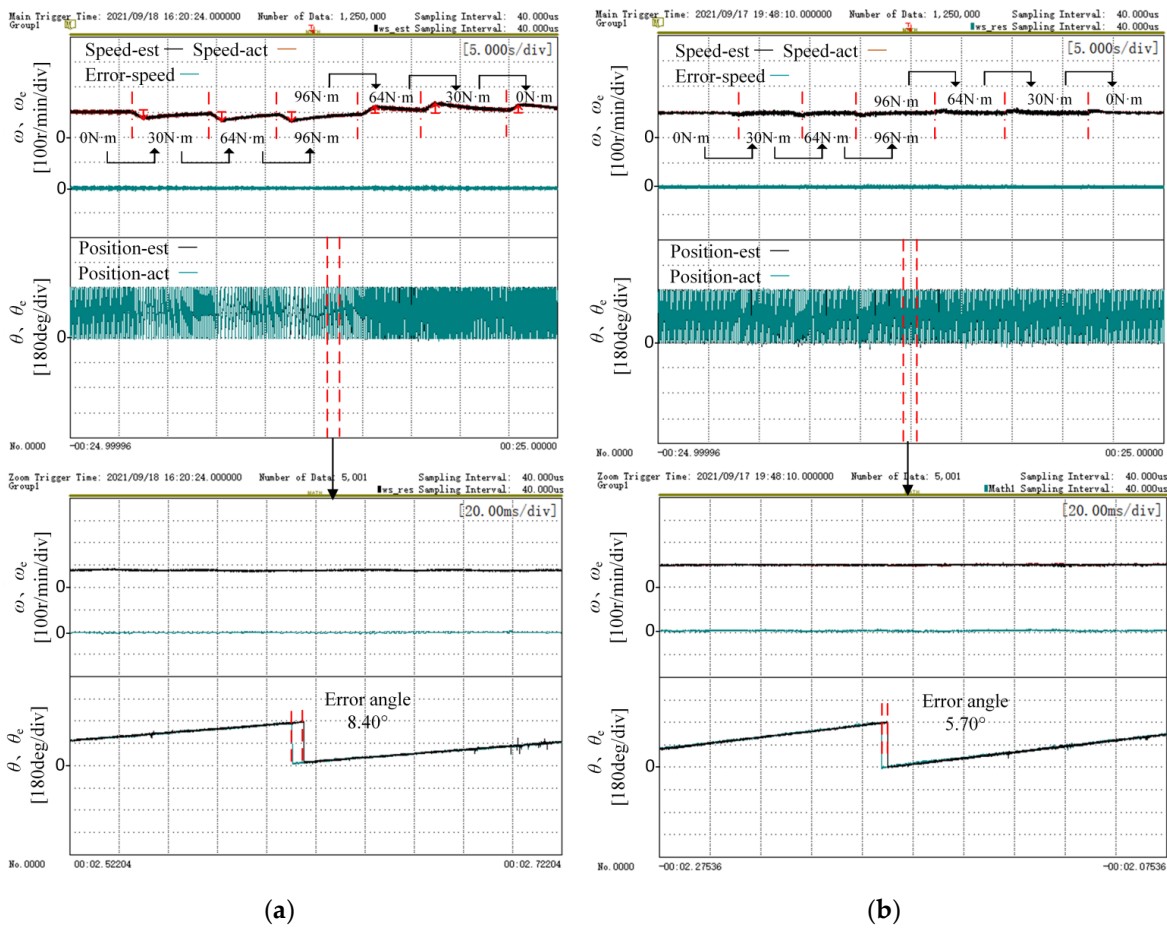

(**a**)                                                                        (**b**)

**Figure 16.** The speed control of tested motor with traditional method and proposed method at 100 r/min: (**a**) traditional method; (**b**) the proposed method.

Table 3 shows the qualitative and quantitative features of the traditional method and the proposed method. It can be seen from the table that the position angle error of the

improved method is about half that of the traditional method, which greatly improves the accuracy of position angle estimation.

**Table 3.** Results discussion.

| | Load Start and Brake Experiment at Low Speed | | Step Load Experiment at Constant Speed | |
| --- | --- | --- | --- | --- |
| | **Quantitative Analysis: (Angle Error)** | **Qualitative Analysis** | **Quantitative Analysis: (Angle Error)** | **Qualitative Analysis:** |
| The Traditional Method | Acceleration: 5.70° 400 r/min: 2.85° Deceleration: 5.54° | Estimated position angle error is large. | When the torque is 96 Nm at 400 r/min: 4.80° | Estimated position angle error is large. |
| The Proposed Method | Acceleration: 2.65° 400 r/min: 1.20° Deceleration: 2.44° | Estimated position angle error is small. | When the torque is 96 Nm at 400 r/min: 2.22° | Estimated position angle error is small. |

## 5. Conclusions

This paper proposes a sensorless control method for high-frequency square wave signal injection into permanent magnet synchronous motors based on a current oversampling scheme. Compared with the traditional high-frequency square wave injection method, this method performs current sampling at the beginning and end of the effective vector and obtains the high-frequency current response through current demodulation, thereby calculating the rotor position and speed. This method increases the frequency of calculating the rotor position by oversampling the current, thereby increasing the frequency of updating the position angle, and reducing the time of position update delay. At the same time, sampling at the beginning and end of the effective vector avoids the error effect caused by the non-linear change of the current in the zero-vector time and improves the rotor position accuracy. The experimental results of load start and brake experiment at low speed and step load experiments at a constant speed show that, compared with the traditional method, the position angle error of the proposed method in the transient process and the steady-state process is about 1–5°, which is about half of the position angle error of the traditional method, and the time for the measured motor current to reach the steady state is shorter. By improving the accuracy of the position angle, the speed range is widened, and the load ability is increased. This has good application value and can be used in engineering practice.

**Author Contributions:** Conceptualization, Z.W. and Z.L.; methodology, Z.W. and T.L.; software, Q.G.; validation, Q.G.; formal analysis, Z.W.; investigation, Z.L.; resources, Z.W.; data curation, J.X.; writing—original draft preparation, Q.G.; writing—review and editing, Z.L.; visualization, Q.G.; supervision, J.X.; project administration, Z.W. and W.C.; funding acquisition, Z.W. and W.C. All authors have read and agreed to the published version of the manuscript.

**Funding:** This research was funded by "The National Natural Science Foundation of China, grant number 51977150" and "The Key Program of Tianjin Natural Science Foundation, grant number 20JCZDJC00020".

**Data Availability Statement:** Not applicable.

**Conflicts of Interest:** The authors declare no conflict of interest.

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
