# Peer review of "High-Frequency Square Wave Injection Sensorless Control Method of IPMSM Based on Oversampling Scheme"

_wevj, doi:10.3390/wevj13110217_

Round 1

Reviewer 1 Report

The theoretical overview and mathematical models are fairly described. The case study is conducted on high-power interior permanent magnet synchronous motors, which gives the soundness of gained results. The ability to apply novel sensorless control method based on oversampling scheme is clearly defined. The contribution is based on proven experimental analysis of IPMSM. The paper contains original research results, due to its contribution, justify publishing.

Below are the technical remarks and disadvantages:

1. The paper conclusion is to short. Extend the conclusion to clarification of the outcomes of the research.

2. Grammatical and typing errors should be corrected.

3. The tags in the text align with the instructions for writing the paper.

4. Match the all image tags with the instructions for writing the paper (e.g. Figure 11.)

Author Response

Thank you very much for your suggestion, we have modified the manuscript. The revised content has also been highlighted in yellow.

Reviewer 2 Report

-          In the introduction, please highlight the improvement introduced by your paper with respect to literature; please give a bulleted list;

-          Please improve the literature investigation in the introduction, and the gap that your work fills in the scientific background;

 Please improve the conclusion by highlighting the results obtained.

Author Response

(The authors gave the same response as above.)

Reviewer 3 Report

This article is interesting; however, the authors need to take in consideration the following suggestions before accepting it:

(1)   The introduction needs to be improved by discussing current articles because only a few new articles have been included in it.

(2)   The authors need to compare their results with other proposals reported in the literature recently in order to validate or show that their contribution is important in the subject.

(3)   A new section named “results discussion” needs to be added in order to condense all obtained results. I recommend presenting a table where the qualitative and quantitative features of your proposal and the other proposals reviewed in order to show the main advantages and disadvantages of your proposal.

(4)   The conclusion needs to be improved, adding quantitative results not only qualitative results. In addition, it is important to mention, what is the next with the investigation? 

Author Response

(The authors gave the same response as above.)

Round 2

Reviewer 2 Report

The paper has been improved in the introduction and conclusions especially.

You could add some more references in the new part of the introduction. 

For example in the new beginning part of the introduction, when you talk about electric vehicle spreading and PMSM torque and speed  measurements make also reference to papers:

De Santis, M.; Agnelli, S.; Patanè, F.; Giannini, O.; Bella, G. Experimental Study for the Assessment of the Measurement Uncertainty Associated with Electric Powertrain Efficiency Using the Back-to-Back Direct Method. Energies 201811, 3536. https://doi.org/10.3390/en11123536" 

Rajeev Ranjan Kumar, Kumar Alok,"Adoption of electric vehicle: A literature review and prospects for sustainability, Journal of Cleaner Production, Volume 253, 2020, 119911, ISSN 0959-6526, https://doi.org/10.1016/j.jclepro.2019.119911."

Reviewer 3 Report

The authors have addressed correctly the Reviewer's concerns. Hence, i can recommend accepting the article.